# The Capacity of Long-Term In Vitro Proliferation of Acute Myeloid Leukemia Cells Supported Only by Exogenous Cytokines Is Associated with a Patient Subset with Adverse Outcome

**DOI:** 10.3390/cancers11010073

**Published:** 2019-01-10

**Authors:** Annette K. Brenner, Elise Aasebø, Maria Hernandez-Valladares, Frode Selheim, Frode Berven, Ida-Sofie Grønningsæter, Sushma Bartaula-Brevik, Øystein Bruserud

**Affiliations:** 1Department of Medicine, Haukeland University Hospital; 5021 Bergen, Norway; annette.brenner@uib.no (A.K.B.); ida.gronningseter@uib.no (I.-S.G.); 2Section for Hematology, Department of Clinical Science, University of Bergen, 5020 Bergen, Norway; elise.aasebo@uib.no (E.A.); maria.hernandez-valladares@uib.no (M.H.-V.); sushma.bartaula@uib.no (S.B.-B.); 3The Proteomics Unit at the University of Bergen, Department of Biomedicine, University of Bergen, 5020 Bergen, Norway; frode.selheim@uib.no (F.S.); frode.berven@uib.no (F.B.)

**Keywords:** acute myeloid leukemia, clonogenic cells, colony formation, cytokines, differentiation, transcription

## Abstract

Acute myeloid leukemia (AML) is an aggressive malignancy, which is highly heterogeneous with regard to chemosensitivity and biological features. The AML cell population is organized in a hierarchy that is reflected in the in vitro growth characteristics, with only a minority of cells being able to proliferate for more than two weeks. In this study, we investigated the ability of AML stem cells to survive and proliferate in suspension cultures in the presence of exogenous mediators but without supporting non-leukemic cells. We saw that a high number of maintained stem cells (i.e., a large number of clonogenic cells after five weeks of culture) was associated with decreased overall survival for patients receiving intensive chemotherapy; this prognostic impact was also detected in the multivariate/adjusted analysis. Furthermore, the patients with many clonogenic cells presented more frequently with mutations in transcription-related genes, and also showed a higher abundance of proteins involved in transcription at the time of diagnosis. In conclusion, the growth characteristics of the long-term proliferating leukemic stem cells seem to have an independent prognostic impact in human AML, and these characteristics appear to be reflected by the mutational landscape and the proteome of the patients at the time of diagnosis.

## 1. Introduction

Acute myeloid leukemia (AML) is an aggressive malignancy, which mainly affects the elderly. It is characterized by an accumulation of immature myeloblasts in the bone marrow [1]. The backbone of treatment is intensive chemotherapy, potentially combined with stem cell transplantation. Still, overall survival is low due to a high relapse rate, and the high number of elderly or unfit patients, who cannot undergo the intensive and potentially curative treatment [2]. 

Acute myeloid leukemia is a highly heterogeneous malignancy and prognosis, and thus overall survival is correlated with both cytogenetics and specific gene mutations, such as FMS-related tyrosine kinase 3 internal tandem repeats (*Flt3*-ITD; unfavorable) and nucleophosmin (*NPM1*; favorable in absence of a coinciding *Flt3*-ITD) mutations [3]. Morphologically, AML is classified into eight distinct categories according to the French-American-British system (FAB M0-7), which defines at which stage in hematopoiesis maturation arrest occurs [4,5]. Cell differentiation is further reflected by downregulation of the CD34 stem cell marker upon maturation [6].

AML cell populations have a hierarchical organization, and a major part of the cells will undergo spontaneous (i.e., stress-induced) apoptosis during the first 3–5 days of in vitro culture [7,8]. However, for some patients, a subset of cells will still proliferate after 2–8 weeks despite the initial apoptosis [9], and our present long-term cultures correspond to these previous in vitro assays for AML stem cells [7]. Furthermore, AML stem cells can also be assayed by xenografts, which is achievable mainly for cells from patients with high-risk leukemia [10]. Finally, total AML cell populations seem to reflect at least certain characteristics of the leukemic stem cells that are thought to be responsible for chemoresistance and leukemia relapse because gene expression and epigenetic profiles of the overall population are associated with prognosis [11,12]. Based on these observations, our hypothesis is that the ability of long-term in vitro growth in the absence of AML-supporting non-leukemic cells differs between patients. This capacity of long-term proliferation reflects a clinically relevant difference in the cellular ability to survive and proliferate under suboptimal conditions and is probably caused by differences in regulation of proliferation and/or balance between pro- and anti-apoptotic signaling. To identify these mechanisms, we characterized a large group of patients with regard to long-term proliferation. Furthermore, to identify possible molecular mechanisms contributing to stress-resistance (i.e., long-term in vitro survival and proliferation) we also performed transcriptomic and proteomic comparisons of primary AML cells with or without long-term in vitro proliferation. 

## 2. Results

### 2.1. Long-Term In Vitro Proliferation of AML Cells in the Absence of Supportive Non-Leukemic Cells Is Detected Only for a Subset of Patients

Primary AML cells derived from 68 unselected patients (Table 1) were cultured in suspension (six-well culture plates); the culture medium was only supplemented with thrombopoietin (TPO), Flt3 ligand (Flt3L), stem cell factor (SCF), granulocyte-macrophage colony-stimulating factor (GM-CSF), and interleukin 3 (IL-3). After 35 days, the non-adherent cells were harvested and thereafter seeded in methylcellulose medium (MethoCult^TM^ H4434 containing erythropoietin, EPO) and incubated for an additional 14 days before the number of colonies was estimated. Thus, the cells were cultured for a total of 7 weeks, and clonogenic cells could then be detected for 38 of the 68 patients. A minority of colonies of erythroid origin (i.e., presenting with morphological signs of hemoglobinization) were detected for 16 of the 38 patients, whereas two exceptional patients showed only erythroid colonies. Thus, non-erythroid colonies dominated for the large majority of patients. The number of colonies obtained from the parallels of samples containing 25,000 and/or 50,000 seeded viable cells per well were back calculated to the initial 2 × 10^6^ cells, taking into consideration that cells from 20 cultures had been removed during the course of suspension culture in order to prevent overgrowth with cell density exceeding 2 × 10^6^ cells/mL. The number of adjusted colonies showed a considerable variation among patients and ranged from 1.5 up to 35,050 colonies per 2 × 10^6^ initial cells (median number: 324 colonies). 

We estimated the number of viable cells after the initial five weeks of suspension cultures, and this number showed a significant correlation with the capacity of colony formation (Kendall’s tau-b: *r* = 0.431; *p* < 0.0001). We further observed borderline correlation with peripheral blood blast count (Kruskal-Wallis test, *p* = 0.055; data from relapse patients were censored) as the median count increased from 40.8 × 10^9^ blasts/L for cultures with less than 0.5 × 10^6^ viable cells to 105 × 10^9^ blasts/L for cultures containing more than 2.0 × 10^6^ viable cells. We defined a threshold of 200 colonies, corresponding to 0.01% long-term proliferating cells, to divide the patients into groups with few and many colonies, respectively. We did this in order not to overestimate the significance of a few observed colonies, which in case of a high cell number can lead to a rather high adjusted colony number. The group with few colonies then comprised 16 patients with a median of 19 colonies whereas the group with many colonies contained 22 patients with a median of 1367 colonies per 2.0 × 10^6^ seeded cells. The number of viable cells after five weeks in suspension culture varied considerably between the groups with no or few detectable colonies on one side and the group of cultures with >200 colonies on the other side (Table 2). Thus, it appears as if cultures with few colonies have more in common with the cultures without detectable colonies, as compared to the group with more than 0.01% long-term proliferating cells. Using this definition, only 1/30 cultures with less than 0.5 × 10^6^ viable cells showed colony formation as opposed to only 2/14 cultures with more than 2.0 × 10^6^ viable cells that did not form at least 200 colonies (Fisher’s exact test: *p* < 0.0001). 

### 2.2. The Capacity of Cytokine-Supported Long-Term In Vitro Proliferation Is Not Associated with Patient Age, Morphological Differentiation, CD34 Expression, Karyotype, or Molecular Genetic Abnormalities

In agreement with previous results [13], the capacity of long-term AML cell proliferation in the absence of supportive cells was not associated with the differentiation status of the primary AML cells at the time of diagnosis, i.e., cultures with and without this capacity did not differ with regard to morphological signs of differentiation (FAB classification) or expression of the CD34 stem cell marker. Furthermore, apart from cell number colony formation after long-term culture appeared to be independent of common prognostic factors such as patient age, hemoglobin levels, peripheral blood blast count, cytogenetics, *NPM1* mutations, and secondary or relapsed versus de novo AML (an overview of patient details is provided in Appendix A). Only *Flt3*-ITD positive cells showed a non-significant trend towards a higher percentage of clonogenic cells because 20% of cultures with <200 colonies presented with *Flt3*-ITD, as opposed to 45% of cultures with higher colony numbers. Thus, at least in this patient cohort, the capacity of long-term in vitro proliferation in the absence of non-leukemic supportive cells seems to be essentially independent of the genetic abnormalities with an established prognostic impact. 

### 2.3. A high Frequency of AML Cells Showing Long-Term Cytokine-Supported Proliferation Is Associated with Reduced Survival for Patients Receiving Intensive Anti-Leukemic Therapy

We compared the overall survival for patients whose blasts did or did not show long-term cytokine-supported proliferation; this comparison included only relatively young and fit consecutive patients with newly diagnosed AML who had completed their intensive and thus potentially curative induction therapy followed by the planned consolidation treatment. The 18 patients showing long-term proliferation were compared with the 17 patients where long-term proliferation supported by cytokines alone was not detectable. Overall survival did not differ between these two groups of patients and the 2-years survival was 40%. However, when we analyzed the quantitative instead of the qualitative differences in long-term proliferation (i.e., the number of colonies/clonogenic cells after 5 weeks of suspension cultures), we detected that the subset of patients with a high number of clonogenic cells (>200 colonies) showed decreased overall survival (median overall survival for patients with no/few detectable colonies being 2.4 years as compared to 8 months for patients with many colonies; log-rank test: 0.006; Figure 1). 

We also performed a multivariate analysis including the known prognostic parameters age (younger or older than 60 years), *Flt3*-ITD, *NPM1* insertions, favorable and adverse/intermediate cytogenetics and disease etiology (secondary versus de novo AML) in addition to the number of colonies (below or above 200 colonies) (Table 3). In the Cox regression analysis, two parameters emerged as independent risk factors for reduced survival: Patient age (hazard ratio, HR = 5.67; *p* = 0.011) and colony number (HR = 5.82; *p* = 0.005). Because the mutation status and/or cytogenetics for four patients were missing (three patients without detected colonies, one with >200 colonies; no long-term survivors), the latter analysis only contained 31 out of the 35 patients with survival data. The lack in associations between prognosis and *Flt3*/*NPM1* mutations is most likely due to the relatively small number of heterogeneous patients in our cohort and a rather large overlap of patients in the groups with *NPM1*-insertions and *Flt3*-ITDs, as 7/18 patients with acquired mutations present with alterations in both genes.

### 2.4. Increased Levels of HGF and IL-1RA during Long-Term Suspension Culture Are Associated with Maintenance of Clonogenic Cells during Culture

We investigated long-term AML cell proliferation in the presence of exogenous cytokines. However, primary AML cells themselves show constitutive release of a wide range of soluble mediators [14,15]. To characterize the mediator release by the leukemic cells during long-term suspension culture we measured the supernatant levels of 19 constitutively released mediators after 7 and 35 days of culture. We first investigated whether the levels of these mediators varied during the five weeks of suspension cultures (Wilcoxon signed-rank test for paired samples; week 5 vs. week 1 values), and whether these variations showed any associations with proliferation/viability during short-term culture, specific AML subgroups (i.e., FAB classification, cytogenetics, gene mutations, disease etiology) or colony formation. First, five of the 19 mediators showed no significant variation (due to a large number of comparisons, the significance level was set to 0.01) during the culture period: the chemokines (CCL5 and CXCL5), the interleukins (IL-1β and IL-6), and granulocyte colony-stimulating factor (G-CSF). Second, the levels of the eight mediators CCL2-4, CXCL8, IL-1 receptor antagonist (IL-1RA), matrix metalloproteases (MMP) 1 and 2, and cystatin C significantly increased during the suspension culture for all patients, except for those without detectable cytokine-dependent proliferation in the one-week ^3^H-thymidine assay and/or extensive spontaneous in vitro apoptosis characterized by less than 30% viable cells after two days of suspension culture. A gradual increase during long-term suspension culture of these eight cytokines thus seems to reflect that viable cells are still present during culture. Third, an increase during culture for the five mediators CXCL1/10, MMP-9, serpin E1 and tumor necrosis factor α (TNFα) was especially seen for patient cells without monocytic differentiation (i.e., FAB M0-2; Appendix A). Finally, an increase in hepatocyte growth factor (HGF) was associated with cultures containing >0.5 × 10^6^ viable cells after five weeks of suspension culture, and with colony forming cultures (Figure 2). 

On the other hand, when we compared the cytokine secretion ratios between different patient subgroups (Mann-Whitney *U*-test; week 5/week 1 ratios; *p* ≤ 0.01), different patterns were observed for seven mediators: CCL2, CCL3, HGF, IL-1RA, MMP-9, cystatin C, and TNFα. Higher ratios of CCL2, CCL3, and cystatin C were observed for cells without *NPM1* insertions and for CD34^+^ cells (Appendix A). Furthermore, higher secretion ratios of IL-1RA, MMP-9 and TNFα were associated with FAB M0-2 (Appendix A). On the other hand, the MMP-9 decrease over time was linked with cells showing morphological changes (i.e., plastic adherence, increased cellular size and/or presence of pseudopodia) over time (*p* < 0.001). Finally, higher ratios of HGF (*p* = 0.004) and borderline of IL-1RA (*p* = 0.014) were observed for cultures with colony forming cells (Figure 3; Appendix A). However, the increase in HGF was most pronounced for the patient group with few colonies. The release ratios for the latter cytokines showed also a weakly positive correlation with the number of colonies: IL-1RA (Kendall’s tau-b: *r* = 0.203; *p* = 0.029) and HGF (*r* = 0.251; *p* = 0.007). 

### 2.5. High Constitutive Cytokine Release Is Associated with an Altered AML Cell Phenotype during the Initial Five Weeks of Suspension Culture 

We reproduced the results from a previous study of primary AML cells [14], which showed that high constitutive release of a wide range of soluble mediators during short-term in vitro culture is associated with signs of monocytic differentiation (FAB M4/5; Appendix A) and the favorable inv(16) abnormality with especially high secretion levels at week 1 and 5, respectively. Also in concurrence with the former study, *NPM1*-insertions and/or absence of the stem cell marker CD34 were associated with lower mediator levels. However, according to the current study, the AML cell population showing morphological alterations after 2–3 weeks of suspension culture had the highest absolute mediator levels of all samples (Mann-Whitney *U*-test; *p* ≤ 0.01): initial (week 1) high levels for CCL4, CXCL8, IL-1RA, HGF, MMP-1/9, and cystatin C; and persistent (week 1 and week 5) high levels for CCL2, CXCL5/10, IL-1β, G-CSF, and TNFα (Appendix A). The presence of morphological changes was associated with monocytic differentiation of the primary cells (FAB M4/5; Fisher’s exact test; *p* = 0.011). All cells with the inv(16)/t(16;16) abnormalities also showed a change in their phenotype, high cytokine levels and in general absence of colony formation, with exception of the single t(16;16) patient (396 colonies). Interestingly, there was, in general, no distinct correlation between constitutive cytokine secretion levels and the number of viable cells after five weeks in suspension culture or colony number after seven weeks of culture (Appendix A). Long-term maintenance seems rather to be associated with a higher relative increase in HGF and probably also in IL-1RA during culture (Section 2.4).

We further performed an unsupervised clustering (Figure 4) including the 14 mediators, which were secreted at significantly higher levels in those cell cultures that showed morphological alterations during long-term culture. In the cluster, the patients divided into two groups: Low and high initial cytokine release. As expected from previous results [14], all five patients with inv(16)/t(16;16) and a majority of 72% of FAB M4/5 patients (Fisher’s exact test; *p* = 0.005) clustered in the high release group. Furthermore, 73% of cultures showing morphological signs of differentiation (increased cellular size and/or presence of pseudopodia) after 5 weeks in suspension clustered in the high release group (Fisher’s exact test; *p* = 0.001). Regarding colony formation, there was no significant difference between the two groups. Intriguingly, morphological alterations during suspension culture appeared to correlate with increased patient survival, independent of colony number (log-rank test *p* = 0.006; Appendix A).

### 2.6. Gene Expression and Mutation Profiling Identifies Differences between Patient Groups with High and Low Numbers of Clonogenic Cells

Using JExpress, we performed a feature subset selection (FSS)/Anova microarray analysis comparing 10 unselected patients with many (>200 colonies) colony forming units (CFUs) with 17 patients with fewer colonies. We set the cut-off to an absolute t-score of 4.00, corresponding to *p* < 5.0 × 10^−4^. We found 14 genes (including two probes for *C3orf38*) differentially expressed, thereof 12 mapped genes. Five of these genes were overexpressed in cultures with <200 colonies (Figure 5; Table 4; Appendix A). Most of the genes are associated with metabolism and energy pathways (*NUDT19*, *P2RX4,* and *TMLHE*), whereas *C3orf38* is possibly linked with apoptosis. The seven upregulated genes for cultures with many colonies, on the other hand, include genes that are linked with mitosis (*KIAA1383* and *PARD6A*), invasion and metastasis (*LMLN* and *PARD6A*), and protein folding (*HSPA12A*). The most interesting gene, however, might be the proto-oncogene *GRB10* (coding for growth factor receptor-bound protein 10) which has been investigated in AML, where it was linked to leukemogenesis [16] and relapse due to resistant disease [17].

Furthermore, for 35 patients—thereof eight with >200 colonies—we have more in-depth information about the mutational landscape including 35 additional mutations to *Flt3*-ITD and *NPM1*-insertions, which all are associated with AML (Appendix A). It appears as if mutations in genes, where the products are involved in chromatid separation (i.e., the chromatid modifier ASXL1 and subunits of the cohesin complex, namely RAD21, SMC1A and STAG2), are more frequent among patients with >0.01% clonogenic cells (0.75 as compared to 0.15 mutations/patient; Fisher’s exact test *p* = 0.010). 

### 2.7. Gene Expression Profiling Identifies Differentially Expressed Genes for Both Cultures Which Present with an Altered AML Phenotype and Cultures with Erythroid Colonies

We performed FSS analysis as described in Section 2.6 for cultures that differed in their AML blast phenotype, and for cultures with and without colonies of erythroid origin.

First, FSS identified 61 probes coding for 35 annotated genes which were differentially expressed between the patient groups with (10 patients) and without (17 patients) stable AML cell phenotype during long-term suspension culture (Section 2.5; Appendix A). The encoded proteins fulfil a wide range of functions with signaling (7 genes), transcription (8 genes) and metabolism (10 genes) being the most prominent (Appendix A). Five of these genes have been studied in AML (see [16,18,19,20,21,22]). 

Second, FSS identified 36 differentially expressed probes, thereof 23 annotated genes (Section 2.1; Appendix A), which differentiated the cultures with erythroid colonies (7 patients) from those containing only myeloid colonies (9 patients). Most of the genes were upregulated in the subset with erythroid colonies and coded for proteins involved in metabolism (5 proteins) and transcription (4 proteins; Appendix A). Two of the upregulated genes have been investigated in AML (see [23,24,25]).

### 2.8. Proteomics Analysis Identifies Proteins Involved in Transcription to Be More Abundant for Patients with a High Number of Clonogenic Cells

We performed both proteomic and phosphoproteomic analyses on cells from 15 consecutive AML patients (8 with no/few and 7 with many colonies) obtained at diagnosis. Thirty-three proteins, thereof 31 with an identified function, showed to be differentially abundant in the proteomic analysis. Transcription factors and proteins involved in mRNA splicing dominated for the patient group >200 colonies, whereas proteins involved in protein synthesis and modification (kinases) were found to be decreased (Table 5; Appendix A). Two of the proteins had been studied in AML previously: VIM (vimentin) and SMG1 (nonsense-mediated mRNA decay associated PI3K related kinase). Vimentin (increased for the group >200 colonies) is considered a negative prognostic factor in AML [26], whereas SMG1 (decreased for the group >200 colonies) is a potential tumor suppressor in hypoxic tumors and associated with regulation of AML cell proliferation [27].

Phosphoproteomic analysis revealed 49 differentially phosphorylated proteins between the two groups, including three proteins that were already identified by the proteomic analysis: carboxypeptidase D (CPD), MAP7 domain containing 1 (MAP7D1) and VIM (Table 5; Appendix A). Again, proteins involved in transcription, translation, and protein-modifying (19 proteins) dominated, in addition to proteins involved in cytoskeleton structure and signaling (7 proteins). As many as 12 proteins had been studied in AML previously. Expression of three of the proteins is linked with mixed-lineage leukemia (MLL/11q23) rearranged AML: The transcription factors DOT1L and ZNF521, and VIM. Furthermore, expression of four proteins is associated with tumor progression/high risk AML: DOT1L, mTOR, VIM and ZNF521 [26,27,28,29,30], whereas the expression of six proteins is associated with tumor suppression/low risk AML: AEBP2, CAST, CBFB, LSP1, MAP1S and probably PCBP1 [31,32,33,34,35]. Finally, the effect of de/phosphorylation has been studied for LEO homolog 1 (LEO1), and dephosphorylation of this protein was then linked to leukemogenesis [36].

## 3. Discussion

In AML, the leukemic cells are organized in a hierarchy [10] and can be divided into three different subgroups with decreasing percentage of the total amount of cells: (i) The non-cycling blasts, which usually undergo apoptosis in cell culture during the first week; (ii) a subset of cells with pro-longed self-renewal capacity (clonogenic cells); and (iii) a small subset of cells with a long-term proliferative capability (4–8 weeks) in cell culture [7,8,9]. The latter include those cells that are referred to as the leukemic stem cells (LSCs) that are usually associated with the CD34^+^CD38^−^ cell subpopulation [37]. Even though maturation is usually associated with loss of CD34 expression and absence of long-term proliferation [8,9], exceptional patients present with CD34^−^ LSCs [9,38,39]. Altogether, these observations fit with our present as well as previous studies [13] where colony formation ability could neither be linked with FAB classification nor CD34 expression. Because the fraction of cells with clonogenic potential is low (median: 1.6 × 10^−2^; mean: 0.16%) these cells can virtually be present in any AML subtype. Since clonogenic blasts in certain cases even exist within the more mature CD34^−^ cell fraction, the cell marker CD34 itself can be used as a predictor neither for colony formation [39] nor for patient survival [37].

One might debate, whether the observed colonies indeed are derived from long-term proliferating leukemic cells or whether they result from contamination with normal hematopoietic stem cells (HSCs). The latter view might be supported by the presence of erythroid colonies for some patients. However, we regard the CFUs to be of leukemic origin. First, a previous study suggested that contamination with healthy HSCs is common [40], and these cells may then proliferate in response to EPO and IL-3, which are present in the MethoCult medium [41,42,43]. Therefore, one would expect to detect erythroid CFUs for most cultures, but only 26% of patients presented with erythroid colonies after 7 weeks and these colonies dominated only for five patients. Second, microarray analysis identified 36 differentially expressed genes (Appendix A) between primary cell populations with versus without erythroid colonies. The different expression patterns between these two patient subgroups suggest that the observed difference lies within the leukemic cell populations, which give rise to different colony types.

We saw an inverse correlation between an increased percentage of clonogenic cells (>0.01%) and patient survival (Figure 1), which appears to be reasonable as these cells are expected to maintain the disease. A larger fraction of blasts with long-term proliferative capacity thus might lead to increased leukemic blast burden in the patients. Long-term replenishing of non-cycling cells could also be observed in the cultures that contained the highest fractions of clonogenic cells, and these cultures frequently (12/22 cultures with >200 colonies as opposed to 2/46 remaining cultures; Fisher’s exact test, *p* < 0.0001) showed a higher number of viable cells after five weeks in suspension culture than the originally seeded 2.0 × 10^6^. 

Previous studies have shown that AML patients differ in their requirements for proliferation during in vitro culture [44,45,46]. Our present assay for long-term in vitro proliferation has two steps that are based on different culture media; first five weeks of suspension culture followed by two weeks of a methylcellulose-based clonogenic assay. The culture medium is thus different but highly standardized in the two assays. Classification of patients as capable of long-term proliferation thus requires survival of sufficient long-term proliferating cells during the first suspension culture, and the ability to proliferate also in the second assay (weeks six and seven of in vitro culture). The two steps in the long-term proliferation assay can explain why certain patients show discrepancies between the numbers of viable cells after five weeks of suspension culture and the number of colonies in the final clonogenic assay, where cultures presenting with few colonies showed a similar amount of viable cells as cultures without colony formation. 

We used primary AML cells derived from peripheral blood of AML patients with high percentages and/or levels of circulating leukemia cells, and as discussed in detail in previous methodological articles we thus have a selection of patients [7,47]. However, our patients should be regarded as representative with respect to the most important biomarkers for in vivo chemosensitivity in AML, i.e., the generally accepted prognostic factors of this disease [47]. In addition, our study shows similar frequencies of various mutants as previous studies of a general AML cell population have indicated [48]. Despite this, we want to emphasize that our results should be interpreted with care because we cannot exclude the possibility that the cell populations are representative only for this selected subset of patients. The advantage of our methodological strategy is that we used a simple and standardized method for preparation of highly enriched AML cells without induction of functional alteration due to more complex separation methods, e.g., antibody-based separation with ligation of surface molecules [7,49]. Finally, the culture of similar AML cell population without subset selection has also been used in previous studies of leukemic stem cells, i.e., long-term proliferating AML cells in the presence of leukemia-supporting stromal cells (see [7,49]).

Another question is whether cells derived from the peripheral blood are representative for the leukemic cells in the bone marrow microenvironment, which is regarded as the site of initial leukemogenesis. A previous study compared the cell surface molecular profile for paired cell samples derived from bone marrow and peripheral blood of the same patients; in these studies, the cells showed only relatively small quantitative differences for a limited number of adhesion molecules [50]. Furthermore, in a recent study, we also showed that constitutive activation of the PI3K-Akt-mTOR pathway varies considerably among patients but has large similarities when comparing paired blood and bone marrow samples from the same patients [51]. These observations are not surprising when taking into account that blood and bone marrow cells are hierarchically organized and are expected to have the same AML-associated genetic abnormalities. Taken together, these observations suggest that peripheral blood samples are representative for the AML cell population in general, but again we would emphasize that our results should be interpreted carefully due to our analysis of peripheral blood rather than bone marrow cells. 

The constitutive cytokine secretion during long-term culture follows the same pattern for a majority (13/19) of mediators, regardless of AML subclassification or long-term proliferation capacity. Thus, the patient heterogeneity with respect to cytokine levels was to a large degree maintained during culture. The cultures that differed from this pattern showed low proliferation and low viability even during the first 2–8 days, indicating that apoptotic/necrotic cells do not considerably contribute to mediator levels due to leakage through damaged/ruptured cell membranes. An exceptional mediator is MMP-9. Even though the levels at week 5 are still highest for both blasts with monocytic differentiation and with altered morphology, the start median values are divided in half for these groups, whereas the concentrations increase by a 4-fold for FAB M0-2 and cells with a stable phenotype. In addition, cultures with clonogenic cells show a trend towards an increase in MMP-9 concentration over time. Together with HGF and IL-1RA–which significantly increased during long-term culture–these were also the only cytokines where the absolute concentrations after five weeks exceeded those of the cultures that lacked clonogenic cells (Appendix A). 

Increased levels for HGF and IL-1RA for cell cultures containing clonogenic cells have been reported previously [13]. At the detected concentrations (<100 ng/mL), IL-1RA stimulates AML cell proliferation [52]. HGF has been thoroughly studied in AML, and enhanced release has been associated with an adverse prognosis [53]. Inhibition of HGF expression in AML cell lines resulted in increased apoptosis and reduced proliferation [54]. Furthermore, inhibition of the HGF-receptor c-MET resulted in reduced colony formation ability of AML cells [55]. Targeting PI3K and mTOR decreased supernatant levels of both HGF and MMP-9 [56]. MMPs play an important role in tumor invasion and metastasis as they degrade extracellular matrix components [57,58]. Especially MMP-9 is associated with extramedullary infiltration and leukemia invasiveness [57,59]. It seems as if HGF promotes cell cycle progression and blast proliferation, whilst inhibiting apoptosis, and increases the expression of MMP-9 [54]. The increased levels of MMP-9 during pro-longed culturing of cultures containing clonogenic cells might, therefore, be a consequence of the increased HGF levels. However, it appears as if the MMP-9 increase during long-term culture is not a specific feature for cultures containing clonogenic cells. Rather than that, depletion of MMP-9 is correlated with cells that show morphological alterations and may be involved in the change of phenotype.

In our present study, we observed a significant difference in supernatant levels only for a few of the constitutively secreted cytokines when comparing cells with and without the capacity of long-term proliferation. These levels reflect the balance between constitutive release, cytokine receptor binding due to autocrine loops, and possibly degradation due to the release of proteases. However, the role of soluble mediators in leukemogenesis is probably very complex, and we, therefore, emphasize that only experiments based on cytokine knockdown, cytokine neutralization, or specific inhibition of cytokine receptors and their downstream signaling can verify whether a certain cytokine is important for the observed difference in long-term AML cell proliferation. Both receptor-blocking of HGF [55] and the use of IL-1β specific antibodies [60] can reduce colony formation, verifying their involvement in the process. Furthermore, as shown in neutralization experiments, even soluble mediators below detection limit may contribute to cell proliferation [47], although the effects induced by the same mediator may vary vastly among different patients [61,62,63,64]. Finally, autocrine effects of a cytokine can be mediated both on intracellular/endogenous as well as extracellular loops [65]. Taken together, differences in cytokine levels among patients (see Figure 2 and Figure 3) do not necessarily reflect the molecular mechanisms behind differences in long-term in vitro proliferation, whereas even mediators with similar supernatant levels in the two patient groups may contribute to long-term proliferation through differences in responsiveness towards the mediator. However, the observed cytokine differences among patients with and without long-term proliferation (Figure 2 and Figure 3) suggest that these two subsets show additional differences with regard to communication via the local cytokine network (i.e., IL-1/IL-1RA, HGF) with neighboring AML-supporting stromal cells in their common bone marrow microenvironment. 

We further observed that the cells which during the course of five weeks showed morphological alterations had the highest constitutive cytokine levels after short-term (1 week) culture for 14/19 mediators (*p* ≤ 0.01), including IL-1RA, HGF and MMP-9 (Figure 4). The week 1 values were chosen because these early measurements correspond the most to the originally cell cultures as (1) no cells have been removed from the cultures, and (2) morphological alterations are not detectable yet. We also want to stress that the 7-day cytokine profile is very similar to that obtained after only two days of culture: 55/68 of the samples were additionally cultured in StemSpan SFEM^TM^ medium at a concentration of 1.0 × 10^6^ cells/mL for two days. The distribution of the patients into two groups with high and low secretion of the same 14 cytokines showed a large degree of overlap with the cluster presented in Figure 4 (Fisher’s exact test; *p* < 0.001). Cultures containing cells with an altered phenotype, therefore, seem to present with an altered cytokine profile; this may explain why these cultures show significantly increased cytokine levels for 7 mediators at week 5, even though the number of viable cells is lower than in cultures with many clonogenic cells (median: 0.88 × 10^6^ vs. 2.46 × 10^6^ cells, respectively; *p* = 0.001). Thus, the week 1 cytokine profile might predict a differentiation potential of the blasts and thereby a favorable prognosis; this would be in agreement with our previous study where prolonged patient survival was associated with a high constitutive cytokine secretion profile after 2 days in culture [14]. This improved survival might then be due to the leukemic cells’ capability to differentiate upon autocrine cytokine stimulation. However, we have to stress that a high number of long-term proliferating cells and an altered phenotype are not mutually exclusive as 50% of cultures with >200 colonies also showed changes in morphology. Still, morphological changes during suspension culture were associated with increased patient survival in our study (Appendix A). 

There are three reasons for why we chose to follow the change in phenotype by light microscopy/cell morphology. First, tracking changes in morphology by light-microscopy has been characterized in previous studies and should, therefore, be regarded as a reliable way to survey alterations in the phenotype, i.e., a monocyte-macrophage differentiation program [66]. Second, repeated evaluations during culture are allowed. Third, the alternative strategy of studying the expression of a single or a few surface markers may reflect aberrant molecular expression (a well-known phenomenon in acute leukemia) rather than an altered phenotype [67]. Thus, in our opinion, this simple methodological strategy is reliable, even though it does not allow any detailed examination.

We also compared the global gene expression profiles of primary AML cell populations with and without more than 0.01% long-term proliferating cells. Since the fraction of clonogenic cells in the cell population is small, it may not be surprising that few genes were found to be differentially expressed between the two groups. Most of the downregulated genes for cultures >200 colonies are involved in (energy) metabolism, whereas the upregulated genes especially code for proteins, which are involved in cell cycle/mitosis and invasion. Of the latter, increased expression of *GRB10* was associated with *Flt3*-ITD, relapse, increased proliferation, reduced apoptosis and even increased colony formation in short-term cultures (2 + 5 days) [17]. Since long-term culturing of AML blasts is not practical in the clinic, these differentially expressed genes (Figure 5; Table 4) and especially *GRB10* might act as biomarkers in order to identify patients with a high fraction of long-term proliferating cells and, thus, worsened prognosis.

We further found that mutations in the cohesin complex (represented by three of the five proteins) and in *ASXL1* are more frequent in the patient group with many clonogenic cells (Appendix A). ASXL1, together with the cohesin complex, regulates gene expression in normal hematopoietic cells. Mutations in *ASXL1* lead to loss of function and altered expression of cohesin target genes [68], and is therefore correlated with poor prognosis in AML [69]. On the other hand, mutations in cohesin are also associated with loss of function, secondary AML, and poor prognosis [70]. Interestingly, co-mutations in *ASXL1* and cohesin genes also appear to be more frequent in the patient group with more than 0.01% long-term proliferating cells. Thus, the loss of function of several proteins with overlapping function might contribute to increased colony formation ability.

Finally, the proteomic analysis showed that many of the differentially abundant proteins are involved in protein biosynthesis, from mRNA splicing to folding and modification of the nascent protein (Table 5; Appendix A). Among these proteins were the negative prognostic factor VIM and the tumor suppressor SMG1, which were found to be increased for the patients with many and few colony forming cells, respectively. Regarding phosphoproteomics, we identified 49 differentially phosphorylated proteins, of which 11 have been linked to prognosis in AML. Knockdown of two of the proteins (histone methylating DOT1L and transcription factor ZNF521) has even shown to reduce colony formation ability in short-term cultures [28,71]. It is remarkable that 40% of the differentially phosphorylated proteins are involved in protein biosynthesis. We hypothesize that there might be a link between the frequent gene mutations in AML and the aberrant phosphorylation profile in the proteins with a similar function. Only six of the 35 patients with mutation data do not present with mutations in epigenetic modifiers, transcription factors/repressors, spliceosome and cohesin complex (Appendix A). Co-mutations in these gene groups are more frequent in the group with many colonies (2.6 as compared to 1.5 mutations/patient; Fisher’s exact test *p* = 0.035). Therefore, aberrant protein expression in AML cells with clonogenic ability might be reflected at both the gene and the protein level. 

Our hypothesis was that the ability of long-term in vitro growth in the absence of AML-supporting non-leukemic cells differed among patients and had clinical relevance. The obtained results strongly suggest that this is true. However, we also hypothesized that differences in the capacity of long-term in vitro proliferation are due to differences in the regulation of AML cell proliferation and/or the balance between pro- and anti-apoptotic signaling; while this may be true, the molecular mechanisms behind these differences still remain to be resolved. 

## 4. Materials and Methods

### 4.1. Patients and Preparation of AML Cells

The study was approved by the local Ethics Committee (Regional Ethics Committee III, University of Bergen, REK 2017-305, Bergen, Norway) and samples were collected after written informed consent. AML blasts from peripheral blood were derived from 68 unselected patients with a high number and/or percentage of circulating leukemic cells (33 females and 35 males; median age 64 years with range 18–92 years). A majority of 45 patients had de novo AML whereas 7 patients had relapsed AML, and 19 patients had AML secondary to chronic myeloproliferative neoplasia, myelodysplastic syndromes or previous chemotherapy (Table 1). Thirty-five of the patients received potentially curative treatment including induction therapy based on cytarabine plus an anthracycline followed by intensive consolidation treatment.

### 4.2. Preparation of Enriched AML Cells

AML cells were isolated from peripheral blood by density gradient separation (Lymphoprep; Axis-Shield, Oslo, Norway; specific density 1.077 g/mL) and due to our selection of patients with high absolute/relative levels of circulating AML cells the gradient-separated cell populations contained at least 90% leukemia cells. The cells were cryopreserved and stored in liquid nitrogen [62]. All cell samples were isolated, cryopreserved, and thawed based on the same highly standardized methods.

### 4.3. Analyses of AML Cell Viability and Short-Term Proliferation

#### 4.3.1. Analysis of AML Cell Viability

AML cells were cultured for 40 h in Stem Span SFEM^TM^ medium (Stem Cell Technologies; Vancouver, BC, Canada) and cell viability analyzed by flow cytometry using propidium iodide-Annexin V (Tau Technologies BV; Kattendijke, The Netherlands) staining as described previously [14]. 

#### 4.3.2. Proliferation Assay

As described in detail previously [14], AML cells were cultured for one week in serum-free Stem Span in the presence of Flt3L, SCF and GM-CSF, until proliferation was assayed as nuclear ^3^H-thymidine incorporation after 8 days. Detectable proliferation was defined as nuclear incorporation corresponding to >1000 cpm.

### 4.4. Long-Term Suspension Cultures

AML cells were cultured in Endothelial Cell Growth Medium (EGM-2 medium, Lonza; Walkersville, MD, USA), which contains, among others, 2% fetal bovine serum and vascular endothelial growth factor (VEGF) to facilitate rapid proliferation. According to the manufacturer’s descriptions, the medium was further supplemented with 20 ng/mL of TPO, Flt3L, SCF, GM-CSF and IL-3 (Peprotech; Rocky Hill, NJ, USA). The EGM-2 medium has shown to support the long-term proliferation of AML blasts [13].

We initially seeded 2 × 10^6^ AML cells in 5 mL EGM-2 medium in 6-well tissue plates (Nunc; Roskilde, Denmark), and the cells were incubated in suspension cultures for five weeks. Once a week, 1–2 mL of medium or cell suspension—for cultures containing high cell density as judged by light microscopy—was removed and replaced by fresh complete EGM-2 medium. Cell supernatants were harvested after week 1 and week 5 and stored at −80 °C prior to analysis.

### 4.5. Colony Formation Assay

Long-term cultured cells were tested in our colony formation assay after five weeks of suspension culture. The non-adherent AML cells were then collected, centrifuged, and resuspended in 0.5 mL RPMI medium (Sigma Aldrich; St. Louis, MO, USA). The number of viable cells was determined using trypan blue staining (Fluka; St. Louis, MO, USA). Subsequently, duplicates of 2.5 × 10^4^ and, where available, 5.0 × 10^4^ viable AML cells were seeded in 0.5 mL MethoCult^TM^ H4434 classic medium (StemCell Technologies; Vancouver, BC, Canada) containing EPO in flat-bottomed 24-well plates (Nunc). Lower viable cell numbers were seeded for patients with low cell populations after five weeks in culture. After an additional 14 days of in vitro culture in the colony formation assay colonies containing more than 30 cells (i.e., CFUs) were scored using an inverted microscope. Colonies were classified as being of either myeloid or erythroid origin, where the latter showed morphological signs of hemoglobinization. 

### 4.6. Analysis of Soluble Mediator Levels in Cell Culture Supernatants

The sampled cell culture supernatants were analyzed by Luminex (R&D Systems; Minnesota, MN, USA). Excretion levels for following 19 proteins were measured: (i) The chemokines CCL2-5, CXCL1/5/8/10; (ii) the interleukins IL-1β, IL-1RA and IL-6; (iii) the growth factors G-CSF and HGF; (iv) the matrix metalloproteases MMP-1/2/9; (v) the protease inhibitors cystatin C and serpin E1; and (vi) TNFα. 

### 4.7. RNA Preparation and Analysis of Global Gene Expression

RNA preparation, labeling, and microarray hybridization have been described in detail previously [72]. Briefly, the gene expression profiles were analyzed using the Illumia iScan Reader (Illumina Inc., San Diego, CA, USA) for fluorescent detection of biotin-labeled complementary RNA (cRNA). The latter was quality-controlled by both NanoDrop and an Agilent 2100 Bioanalyzer (Agilent Tachnologies, Santa Clara, CA, USA). Finally, 750 ng biotin-labeled cRNA was hybridized to the HumanHT-12 V4 Expression BeadChip (Illumina Inc.), which targets 47,231 probes. The microarray data were quantile normalized prior to analysis in J-Express 2012 (MolMine AS, Bergen, Norway) [73].

### 4.8. Proteomic and Phosphoproteomic Analyses

The methods for analysis of proteomic and phosphoproteomic profiles in primary human AML cells have been described in detail previously [74,75,76]. Briefly, each patient sample was spiked with a heavy super-SILAC mix [77] and prepared according to the filter-aided sample preparation protocol [78] for proteome and phosphoproteome analysis using nano-liquid chromatography online coupled with a QExactive HF Orbitrap mass spectrometer (Thermo Scientific, Bremen, Germany). 

### 4.9. Statistical and Bioinformatical Analyses

The statistical analyses were performed with the IBM Statistical Package for the Social Sciences (SPSS) version 25 (Chicago, IL, USA) and with GraphPad Prism 5 (San Diego, CA, USA). The Wilcoxon signed-ranked test was used to compare paired samples, whereas Mann-Whitney *U*-tests and Kruskal-Wallis tests were used to compare different patient subgroups. The χ^2^ test was used to analyze categorized data and the Kendall’s tau-b test for correlation analyses. Finally, Kaplan Meier analysis and log-rank test in addition to Cox regression for multivariate analyses were used for patient survival statistics. In general, *p*-values < 0.05 were regarded as statistically significant. 

The statistical analyses of proteomic and phosphoproteomic data were performed with a Welch *t*-test in Perseus (Max Planck Institute of Biochemistry, Martinsried, Germany) [79], and the significance of fold changes was assessed using Z-statistics [80].

For hierarchical clustering of the cytokine release levels, all values were median normalized and log transformed prior to clustering using J-Express (MolMine AS, Bergen, Norway). Complete linkage and cosine correlation were used as the linkage method and distance measurement, respectively.

## 5. Conclusions

In this study, more than half of the AML patients had cells with long-term (i.e., seven weeks) proliferative capacity and/or a stable phenotype during culture; both these characteristics are associated with an adverse prognosis for patients receiving intensive induction therapy. Strikingly, these two parameters were not associated with other unfavorable prognostic factors. Cultures containing clonogenic cells presented with a higher increase in HGF and IL-1RA concentrations than the others during culture, and had, in general, higher numbers of viable cells after 5 weeks in suspension culture. From our results, it appears as if aberrant transcription and transcription regulation is associated with a high percentage of clonogenic cells, as genes involved in transcription more frequently are mutated for these patients, and proteins which are involved in protein synthesis are overexpressed or differentially phosphorylated for the patient group. Thus, these characteristics may be used to predict the presence of many clonogenic cells at the time of diagnosis, and thus an adverse prognosis. 

## Figures and Tables

**Figure 1 cancers-11-00073-f001:**
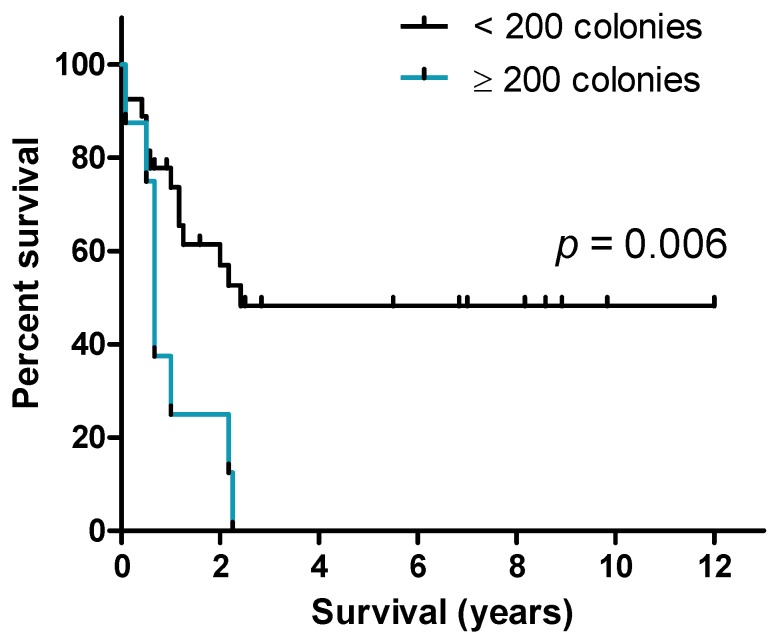
Patient survival dependent on colony number. Patients with blasts giving rise to more than 200 colonies after long-term culture (blue line) showed significantly (log-rank test) worse outcome than the patients with fewer colony forming cells.

**Figure 2 cancers-11-00073-f002:**
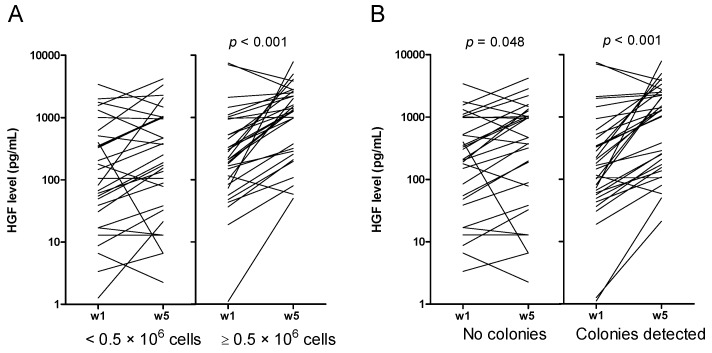
Differences in hepatocyte growth factor (HGF) secretion dependent on cell population and colony number. (**A**) HGF concentrations do not significantly increase between week 1 and week 5 for cultures containing less than 0.5 × 10^6^ viable cells after 5 weeks of culturing. (**B**) Increase in HGF levels is more pronounced in cultures with detectable colonies (right). Note the large overlap between the patients with higher cell numbers and colony formation.

**Figure 3 cancers-11-00073-f003:**
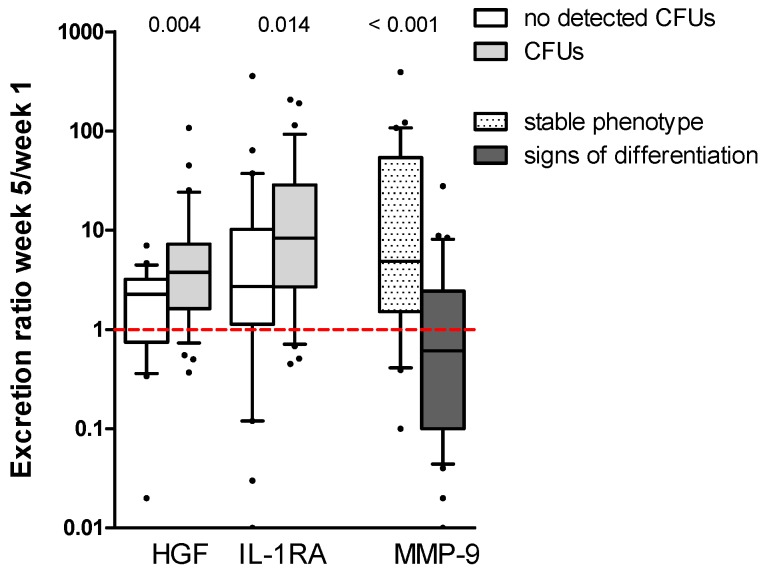
Cytokine secretion week 5/week 1 ratios for the two cytokines that were significantly higher expressed by cultures containing clonogenic cells. Median ratios HGF: 2.26 vs. 3.77; IL-1RA: 2.71 vs. 8.31. The right part of the figure shows that matrix metalloprotease (MMP)-9 is depleted from the medium over time for cultures, where cells present with a change in their phenotype: 4.86 vs. 0.61. CFU: colony forming unit.

**Figure 4 cancers-11-00073-f004:**
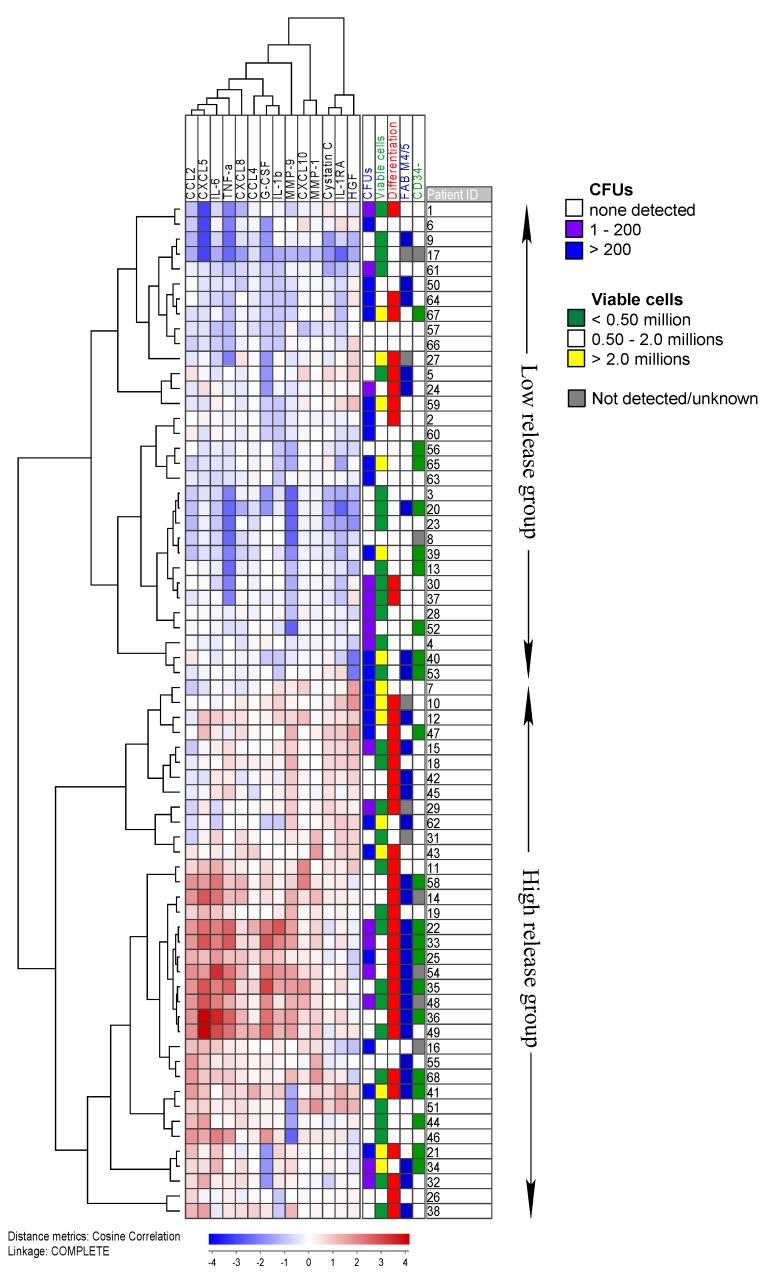
A cluster of the 14 cytokines that were differentially expressed for cells that showed morphological changes during suspension culture. The cluster shows the week 1 concentrations for all 68 patients. All values were median normalized and log transformed, thus the blue color indicates values below the median. Information about the number of detected colonies, the viable cell population, the presence of morphological changes over time and the blast morphology, according to the French-American-British (FAB) system and CD34 expression, is provided on the right. Both blasts with a change in their phenotype during culture and with signs of monocytic differentiation (FAB M4/5) are overrepresented in the high release group. On the other hand, colony number correlates with cell number but not with cytokine expression.

**Figure 5 cancers-11-00073-f005:**
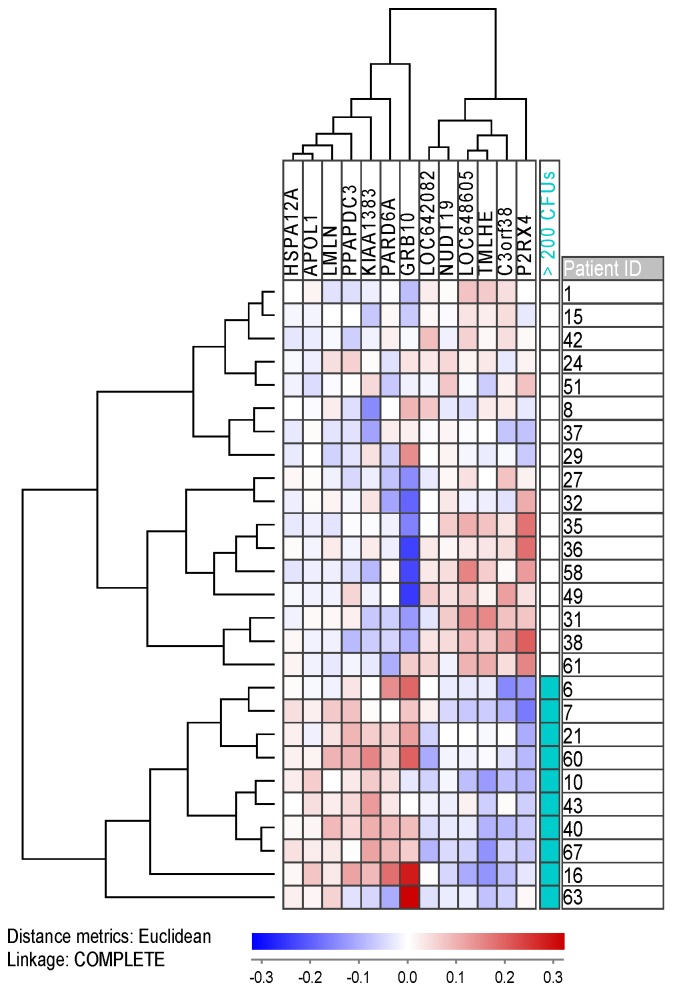
A cluster of the 13 differentially expressed genes between cultures with many (>200) and few colony forming cells. Probes of differentially expressed mRNA with an absolute *t*-score > 4.00 were median normalized and log transformed prior to clustering. *C3orf38* represents the mean value for the two gene probes. Red and blue color represent up- and downregulated genes, respectively. *RPS4Y1*, a gene encoded on the Y-chromosome, was excluded from the cluster as it divides the patients by gender rather than colony number; the patient subgroups presented by the microarray coincidentally contained 70% female patients in the high colony subgroup, and 82% male patients in the low colony subgroup.

**Table 1 cancers-11-00073-t001:** Biological and clinical characteristics of the 68 acute myeloid leukemia (AML) patients included in the study.

Patient Characteristics, Cell Morphology		Disease Etiology and Cell Morphology		Cell Genetics	
*Age*		*De novo AML*	45	*Cytogenetics*	
Median (yrs)	64			Favorable	7
Range (yrs)	18–92	*Secondary AML*		Intermediate	5
		MDS	10	Normal	35
*Gender*		CMML	2	Adverse	15
Females	33	CML	1	n.d.	6
Males	35	CLL	1		
		MF	3	*Flt3 mutations*	
*FAB classification*		PV	1	ITD ^2^	16
M0	4	Chemotherapy	2	Wild-type ^2^	39
M1	17			n.d.	13
M2	13	*AML relapse* ^1^	7		
M4	15			*NPM1 mutations*	
M5	14	*CD34 receptor*		Mutated	20
n.d.	5	Negative (≤20%)	21	Wild-type	33
		Positive (>20%)	41	n.d.	15
		n.d.	6		
				*Flt3 plus NPM1 mutation*	9

^1^ Three of the patients had a relapse of secondary AML. ^2^ One patient in each group has a point mutation at D835. n.d.: not determined. Abbreviations: MDS: myelodysplastic syndrome; CMML: chronic myelomonocytic leukemia; CML: chronic myeloid leukemia; CLL: chronic lymphocytic leukemia; MF: myelofibrosis; PV: polycytemia vera; ITD: internal tandem repeat.

**Table 2 cancers-11-00073-t002:** Overview of median and range values for colony number and cell population for the groups without detectable colonies, with few colonies and with many colonies.

	Number of Colonies		
	Not Observed	1–200	>200
Colony number	0	19 (1.5–180)	1367 (285–35,050)
Viable cell number (10^6^)	0.26 (0–3.06)	0.18 (0.01–2.25)	2.46 (0.30–53.5)

**Table 3 cancers-11-00073-t003:** Multivariate (adjusted) hazard ratio analysis for parameters associated with prognosis in AML. Significant *p*-values are highlighted.

Variable	Adj. HR	95% CI	*p*-Value
Age (≥60 years) ^1^	5.67	1.49–21.56	0.011
Colony number (≥200)	5.82	1.72–19.66	0.005
*Flt3*-ITD	1.83	0.48–6.92	0.377
*NPM1*-insertion	2.33	0.65–8.39	0.194
Etiology (*de novo*)	0.44	0.05–3.89	0.458
Favorable cytogenetics	0.78	0.11–5.61	0.802
Unfavorable/intermediate cytogenetics	0.58	0.11–3.00	0.513

^1^ Reference value, i.e., age ≥ 60 years is a risk factor. HR: hazard ratio; CI: confidence interval.

**Table 4 cancers-11-00073-t004:** Classification of the genes that were found to be differentially expressed for patient populations with >0.01% clonogenic cells. The upregulated genes are associated with cell cycle and signaling, whereas the downregulated genes mostly are linked with metabolism. For details on the encoded proteins, see Appendix A.

Classification Based on Protein Function	Gene Expression in AML Population with Colony Formation	
	Upregulated	Downregulated
Cell cycle, mitosis	*KIAA1383*/*MTR120*; *LMLN*; *PARD6A*; *GRB10*	
Transcription		(*RPS4Y1*)
Apoptosis		*C3orf38*?
Chaperone	*HSPA12A*	
Signaling, transport	*APOL1*; *GRB10*; *PPADDC3*/*PLPP7*	*P2XR4*; *TMLHE*
Metabolism		*NUDT19*; *TMLHE*
Unknown		*LOC642082*; *LOC648605*

**Table 5 cancers-11-00073-t005:** Classification of the proteins that were found to be differentially abundant (left) or differentially phosphorylated (right) for patient populations with >0.01% clonogenic cells. The proteins with higher abundance are associated with transcription and lipid/ion transport. The differentially phosphorylated proteins, on the other hand, are mainly linked with protein biosynthesis and the cytoskeleton. For details on the proteins, see Appendix A.

Biological Process	Proteomic Analysis		Phosphoproteomic Analysis	
	Increased ≥200 Colonies	Decreased ≥200 Colonies	Increased ≥200 Colonies	Decreased ≥200 Colonies
mRNA splicing	RBMXL1; TXNL4A; ZNF830		PCBP1 ^2^	PAPOLA; PRPF31; PRPF40A
Transcription	MED1; POLR3C; ZNF830	SMG1 ^2^	DOT1L ^2^; TCEAL3; ZNF521 ^2^	CBFB ^2^; ISL2; LEO1 ^2^; MYSM1
Transcription repressor			SPEN; ZNF521 ^2^	AEBP2 ^2^; GATAD2B
Translation		ABCF1; EIF2B2; FARSB; MRPL3		DNAJC2; EIF5B
Kinase		CAMK2D; DGKZ; SMG1 ^2^		NEK1; PRKD2
Chaperone				DNAJC2; DNAJC5
Metabolism	CPD ^1^; HEXB; MPDU1	USP4	CAST ^2^; CPD ^1^; DOT1L ^2^	CTPS1; MYO9B; PPP6R3; RASGRP2
Transporter, ion channel	CLIC4; CPD ^1^; CRIP1; RALGAPA2	ABCF1; CLCN3; RBP4	CPD	ATP13A1; OSBP; SCAP
Signaling	RALGAPA2; VIM ^1,2^	CAP1; MRC2	VIM ^1,2^; ZFYVE26	LAT ^2^; LSP1^2^; mTOR ^2^; RNF31; TMPO
Cytoskeleton	VIM ^1,2^	AAMP; CAP1; MYOF	SP100; VIM ^1,2^; ZFYVE26	DBNL; MYO9B; SH3BP1; TMPO
Apoptosis	MED1; RMDN3; TMEM173			GRAMD4; MAP1S ^2^
Cell cycle/mitosis			KIF4A	NEK1
Proliferation regulation				FAM53C; SCRIB
Others	C19orf19/TRIR; MOB4			BOD1L1; CAD; IRF2BP2; TTC7A
Unknown	GPALPP1	MAP7D1 ^1^		MAP7D1 ^1^; TOR4A

^1^ These proteins are found in both proteomic and phosphoproteomic analysis. ^2^ These proteins have been studied in AML.

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
