# Peer review of "The Capacity of Long-Term In Vitro Proliferation of Acute Myeloid Leukemia Cells Supported Only by Exogenous Cytokines Is Associated with a Patient Subset with Adverse Outcome"

_cancers, 2019, doi:10.3390/cancers11010073_

Round 1

Reviewer 1 Report

In this study, Dr. Brenner and colleagues investigated the correlation between long-term in vitro proliferation of acute myeloid leukemia cells in the absence of AML-supporting non-leukemic cells and patient’s survival.  As well as deeper analysis of differences between long-term and short-term in vitro AML cells proliferation and low /high colony formation were addressed. Several up to date approaches were used in this study, specifically measurement of mediators released by AML cells, gene expression, and proteomic analysis.

To analyze the differences between long-term and short-term in vitro proliferation of acute myeloid leukemia cells is a very relevant translational goal, which can result in identification of potential prognostic markers for AML patient. However, some questions are arising in this manuscript, which should be addressed.

(i)                  Line 73: comma should be added after (GM-CSF)

(ii)                Table 2: Can the authors explain or suggest what may be the reason why higher number of viable cells (0.26) after 5 weeks of incubation resulted in a lower number of colonies (0) compared to the lower number of viable cells (0.18), which resulted in 1-200 colonies? Logically, it will be predicted that more viable cells will be able to create more colonies.

(iii)              Figure 1 legend: The survival analysis is based on two groups comparison, (1) patients with <200 colonies and (2) patients with ≥ 200 colonies. However, in the figure legend the group 2 (patients with 200 colonies) is addressed as patients with more than 0.01% long-term proliferating cells. This may be confusing for readers. This comparison is based on colonies numbers and not numbers of long-term proliferating cells even though there is possible correlation between long-term proliferating cells and number of colonies. I suggest using following classification of these two groups: (1) patients with <200 colonies and (2) patients with ≥ 200 colonies.

(iv)              Table 3: Can the authors explain why there was no significant correlation between patient’s survival and Flt3/NMP1 mutation, when in introduction part it is mentioned that overall survival usually correlates with this specific mutation (Line: 40-43)?

(v)                Line 169: The abbreviation for IL-1 receptor antagonist should be referred in the brackets as (IL-1RA). IL-1 receptor antagonist (RA) is confusing.

(vi)              The whole manuscript: authors should consider replacement of cytokine excretion for cytokine secretion

(vii)            Line 230-231: The orientation of two groups of patients in the Figure 4 (above, below) should be removed. It may be confusing in the text as the above notification is pared with low initial cytokine release and the below with high cytokine release. There is no need for these additional notes for orientation in Figure 5.

(viii)          Figure 5: the figure is named as “Cluster of the 14 differentially expressed genes…”. However, in the actual figure only 13 of these genes are presented. Even thought, the explanation is clarified later (RPS4Y1 was excluded), the authors should refer to this figure as “Cluster of 13 differentially expressed genes…” to eliminate the confusion.  

(ix)              In Figure 4. Authors are referring to the completely left column as the Patient ID. However, in the Figure 5 it is referred just as an ID. The authors should unify this.

Overall the manuscript is well written, the experiments are designed logically, and the results can have a valuable impact on the studied field. I also appreciate the conclusion part, where the authors showed their critical thinking as they agree that long-term culturing of AML blasts cannot be easily translated into the clinical practice. Therefore, different approaches for identification of possible prognostic markers in AML patients were presented.

Author Response

1.1 In this study, Dr. Brenner and colleagues investigated the correlation between long-term in vitro proliferation of acute myeloid leukemia cells in the absence of AML-supporting non-leukemic cells and patient’s survival.  As well as deeper analysis of differences between long-term and short-term in vitro AML cells proliferation and low /high colony formation were addressed. Several up to date approaches were used in this study, specifically measurement of mediators released by AML cells, gene expression, and proteomic analysis.

To analyze the differences between long-term and short-term in vitro proliferation of acute myeloid leukemia cells is a very relevant translational goal, which can result in identification of potential prognostic markers for AML patient. However, some questions are arising in this manuscript, which should be addressed.

Line 73: comma should be added after (GM-CSF)

The comma was added.

1.2 Table 2: Can the authors explain or suggest what may be the reason why higher number of viable cells (0.26) after 5 weeks of incubation resulted in a lower number of colonies (0) compared to the lower number of viable cells (0.18), which resulted in 1-200 colonies? Logically, it will be predicted that more viable cells will be able to create more colonies.

There will always remain uncertainty when determining cell and colony number by light microscope. By dividing the cultures into groups with definitive high colony numbers and those with lower numbers, we were able to reduce the gray zone that we would have using a strict yes-no approach. Because we could not seed all the cells after 5 weeks, for patients with high cell numbers but relatively few colony forming cells, coincidental presence or absence of single colony forming cells in a subset of the patient`s cells makes the pure yes-no approach arbitrary. By defining a threshold, such cultures will end up in the same group of no/few colonies. The results regarding cell numbers after 5 weeks in culture and patient survival reflect that the group of patients with many colonies is biologically different. We stressed this approach by adding another sentence in section 2.1 (lines 99-102).

Regarding why the cultures without colonies show somewhat higher cell numbers than the group with few colonies, this might be due to that cell number was assessed two weeks before the end of the experiment. After 5 weeks we transferred the cells to another medium, and blasts derived from different patients might react differently to this change. Therefore, the cell numbers after 7 weeks are likely to differ from those after 5 weeks. We added a new paragraph about this topic in the discussion (lines 374-384). However, we still would like to emphasize that there in general was a high correlation between viable cells after 5 weeks and colony number after two additional weeks of culture in the clonogenic assay, and that the 25-percentile of viable cell numbers for the two groups with no or few colonies was identical; thus, these two groups appear to behave quite similar.

1.3 Figure 1 legend: The survival analysis is based on two groups comparison, (1) patients with <200 colonies and (2) patients with ≥ 200 colonies. However, in the figure legend the group 2 (patients with 200 colonies) is addressed as patients with more than 0.01% long-term proliferating cells. This may be confusing for readers. This comparison is based on colonies numbers and not numbers of long-term proliferating cells even though there is possible correlation between long-term proliferating cells and number of colonies. I suggest using following classification of these two groups: (1) patients with <200 colonies and (2) patients with ≥ 200 colonies.

We changed the sentence according to the reviewer`s suggestion (page 4).

1.4 Table 3: Can the authors explain why there was no significant correlation between patient’s survival and Flt3/NMP1 mutation, when in introduction part it is mentioned that overall survival usually correlates with this specific mutation (Line: 40-43)?

We agree that Flt3 and NPM1 are established prognostic parameters in human AML. Despite this they did not reach statistical significance in our statistical analyses, and we think this is due to two reasons: first, we have relatively few patients so that patients that fit within these groups will be heterogeneous with regard to other factors. Second, in the survival analysis as many as 7 patients showed mutations in both genes. Thus, these patients made up at least 50% of the patients in each of the subgroups (Flt3-ITD or NPM1-ins). We commented on that in section 2.3 (lines 154-158). See also the answer to the first comment of reviewer 2.

Reviewer 2> However, the manuscript flow is difficult to follow and some conclusions are too strong. For example: “the fraction of clonogenic cells was not correlated with other unfavorable prognostic factors, such as adverse cytogenetics, disease relapse, secondary AML or patient age”: given the high variability in the AML molecular landscape, it is likely that no significance comes up in this experimental design due to limited number of cases. It is hard to believe thatadverse outcome to therapy correlates with clonogenic potential but does not with molecular/cytogenetic prognostic factors, as supported by manuscript of hundreds/thousands of patients.

AuthorsWe agree Flt3 and NPM1 mutations are established prognostic factors in AML. The reason why they do not reach statistical significance in our analyses is probably that we have investigated a relatively small group of patients. Our patients have the expected heterogeneity when investigating unselected

4

patients, and several patients have both Flt3 and NPM1 mutations. This is now commented in section 2.3 (lines 154-158).

1.5 Line 169: The abbreviation for IL-1 receptor antagonist should be referred in the brackets as (IL-1RA). IL-1 receptor antagonist (RA) is confusing.

The abbreviation was changed according to the reviewer`s suggestion.

1.6 The whole manuscript: authors should consider replacement of cytokine excretion for cytokine secretion

Changes throughout the text and the supplementary were undertaken.

1.7 Line 230-231: The orientation of two groups of patients in the Figure 4 (above, below) should be removed. It may be confusing in the text as the above notification is pared with low initial cytokine release and the below with high cytokine release. There is no need for these additional notes for orientation in Figure 5.

Changes were made according to the reviewer`s suggestion. We also removed it from the figure caption.

1.8 Figure 5: the figure is named as “Cluster of the 14 differentially expressed genes…”. However, in the actual figure only 13 of these genes are presented. Even thought, the explanation is clarified later (RPS4Y1 was excluded), the authors should refer to this figure as “Cluster of 13 differentially expressed genes…” to eliminate the confusion.  

Thank you for pointing that out. We changed the number to 13.

1.9 In Figure 4. Authors are referring to the completely left column as the Patient ID. However, in the Figure 5 it is referred just as an ID. The authors should unify this.

In both figures “patient ID” was selected.

1.10 Overall the manuscript is well written, the experiments are designed logically, and the results can have a valuable impact on the studied field. I also appreciate the conclusion part, where the authors showed their critical thinking as they agree that long-term culturing of AML blasts cannot be easily translated into the clinical practice. Therefore, different approaches for identification of possible prognostic markers in AML patients were presented.

Thank you very much for the summary and the constructive comments to our manuscript.

Reviewer 2 Report

The manuscript by Brenner at al. is of potential interest since in vitro culturing of primary AML blast cells is a challenge and I appreciate the attempt to correlate this feature with the cell molecular landscape and transcriptomic profile.

However, the manuscript flow is difficult to follow and some conclusions are too strong.

For example: “the fraction of clonogenic cells was not correlated with other unfavorable prognostic factors, such as adverse cytogenetics, disease relapse, secondary AML or patient age”: given the high variability in the AML molecular landscape, it is likely that no significance comes up in this experimental design due to limited number of cases. It is hard to believe that adverse outcome to therapy correlates with clonogenic potential but does not with molecular/cytogenetic prognostic factors, as supported by manuscript of hundreds/thousands of patients.

Some paragraphs are not so innovative or report only negative results.

The manuscript should focus on a clearer message supported by the data.

Author Response

2.1 The manuscript by Brenner at al. is of potential interest since in vitro culturing of primary AML blast cells is a challenge and I appreciate the attempt to correlate this feature with the cell molecular landscape and transcriptomic profile.

However, the manuscript flow is difficult to follow and some conclusions are too strong.

For example: “the fraction of clonogenic cells was not correlated with other unfavorable prognostic factors, such as adverse cytogenetics, disease relapse, secondary AML or patient age”: given the high variability in the AML molecular landscape, it is likely that no significance comes up in this experimental design due to limited number of cases. It is hard to believe that adverse outcome to therapy correlates with clonogenic potential but does not with molecular/cytogenetic prognostic factors, as supported by manuscript of hundreds/thousands of patients.

We agree Flt3 and NPM1 mutations are established prognostic factors in AML. The reason why they do not reach statistical significance in our analyses is probably that we have investigated a relatively small group of patients. Our patients have the expected heterogeneity when investigating unselected patients, and several patients have both Flt3 and NPM1 mutations. This is now commented in section 2.3 (lines 154-158).

2.2 Some paragraphs are not so innovative or report only negative results.

This is correct, and due to this comment we have transferred some sections/figures/tables from the article to the Supplementary information; certain parts of the text have also been shortened. The following alterations have been made:

Section 2.5: the correlation between cytokine levels and both cell and colony number was shortened (lines 225-227) and the data are now presented in the new supplementary table S5.

Section 2.7: the discussion of single proteins with association to AML were removed, and the information instead included in the supplementary tables S8 and S10.

The discussion about the proteins determined in section 2.7 were removed entirely from the discussion.

2.3 The manuscript should focus on a clearer message supported by the data.

We hope the alterations described in response 2.2 have also helped us to present our message more clearly. In addition we have carefully controlled our headings, and several of them have been rewritten to better reflect the observations described in the corresponding text.

Reviewer 3 Report

In the abstract and introduction, it needs to be more clearly defined as to why this study is important in advancing the leukemic field. This paper cannot be published without discussing important previous work in this area (for example: Cytokine-Mediated Inflammatory Pathways Promote Clonal Evolution and Disease Progression in Acute Myeloid Leukemia. Christopher O. Eden, David K. Edwards V, Christopher A. Eide, Elie Traer, Jeffrey W. Tyner, Shannon K. McWeeney and Anupriya Agarwal, Blood 2016.)

The experiments are nicely done and show association of makers. It would be nice if these associations could be furthered shown to be necessary. For example, HGF is mentioned as being associated with long term maintenance, but is it necessary, if it’s knocked down do cells survive and proliferate, this should be addressed.

Samples used were from peripheral blood only. Would cells from bone marrow aspirates be different as this is the initial site of leukemic transformation?

Are samples seeded on supporting cells similar to those not seeded on supporting cells?

What is the initial cytokine profile of the samples? It would nice to see this data as this is what is occurring in a patient.

Hypothesis: “The capacity of long-term in vitro growth in the absence of AML-supporting non-leukemic cells is due to a different regulation of proliferation and a balance between pro- and antiapoptotic signaling that favor survival and proliferation even during suboptimal in vitro conditions.”

- Section 2.4: This where an initial cytokine level before culture would be nice. Using the word association is good because there isn’t a direct causation shown but this would be nice to see by potentially blocking the cytokines present. Does this alter the long-term culture ability?

- Section 2.6-2.8: Same as in 2.4.

In your hypothesis, it states that long term in vitro growth is due to “a different regulation of proliferation and a balance between pro- and antiapoptotic signaling that favor survival and proliferation even during suboptimal in vitro conditions” but this is not actually proven. So, the hypothesis wasn't proven and additional experiments are needed. 

Author Response

3.1 In the abstract and introduction, it needs to be more clearly defined as to why this study is important in advancing the leukemic field. This paper cannot be published without discussing important previous work in this area (for example: Cytokine-Mediated Inflammatory Pathways Promote Clonal Evolution and Disease Progression in Acute Myeloid Leukemia. Christopher O. Eden, David K. Edwards V, Christopher A. Eide, Elie Traer, Jeffrey W. Tyner, Shannon K. McWeeney and Anupriya Agarwal, Blood 2016.)

We now include a more thorough discussion about both HGF and other cytokines which might contribute to long-term proliferation/colony formation. The article about the specific role of IL-1 was added (ref. 60). The reason why we did not see a connection between IL-1β and colony formation is probably due to low secretion levels of the cytokine, having a median value of 3 pg/mL in contrast to the used 10 ng/mL in the reference. Also, IL-1RA has shown to have divergent effects on AML cell proliferation (ref. 61), and we do report differences of this interleukin. Thus, IL-1 might play an important role in colony formation. We added a new section about the topic in the discussion (lines 442-462).

3.2 The experiments are nicely done and show association of markers. It would be nice if these associations could be furthered shown to be necessary. For example, HGF is mentioned as being associated with long term maintenance, but is it necessary, if it’s knocked down do cells survive and proliferate, this should be addressed.

It has not been possible for us to perform new experiments within the deadline for submitting the Revised Version. However, we definitely agree with the reviewer that an association does not demonstrate a causal connection; this can only be established in knockdown/neutralization/blocking experiments. This is now clearly stated and discussed in a new chapter of the Discussion section (page 14). The comment included in the article is shorter than the more detailed comment to the reviewer included in this letter.

The question whether constitutive cytokine release is important for the observed differences in long-term proliferation is in our opinion very complex. Previous studies in short-term suspension cultures and clonogenic assays of hematopoietic growth factor effects on in vitro AML cell proliferation have shown that the effect of single growth factors differ among patients, some growth factors may even have divergent effects and single patients often respond to several growth factors. Thus, the proliferative responsiveness profile to a growth factor panel will therefore differ among patients. In our opinion it is therefore most likely that the possible contribution of constitutively released growth factor to the long-term proliferation will differ among patients.

As stated above we would emphasize that only experiments based on cytokine knockdown, cytokine neutralization or specific inhibition of cytokine receptors and their downstream signaling can verify that a certain cytokine is important for or contributes to a difference in AML cell proliferation; a difference in supernatant levels alone is not sufficient to reach such a conclusion even though previous studies have shown that both HGF and IL1 can function as growth regulators for primary human AML cells cultured in suspension cultures (ref. 61). Firstly, neutralization experiments have demonstrated that even soluble mediators not reaching detectable supernatant levels in the supernatants (ref. 47). Secondly, constitutive release of certain cytokines (including HGF and IL-1) may have divergent effects between patients with regard to AML cell proliferation (refs. 61-64). Thirdly, the autocrine effects of a cytokine can be mediated both on intracellular or endogenous as well as extracellular loops (ref. 65). Taken together, these observations suggest that autocrine effects mediated by cytokines that differ among patients are not necessarily important for long-term in vitro proliferation, and mediators that do not differ in their supernatant levels may contribute to long-term proliferation through differences in responsiveness to the mediator. Based on previous studies it seems most likely that the contribution of single cytokines to long-term proliferation will differ among patients, and the proliferation may depend more on the overall cytokine release profile than on single cytokines (refs. 62-64). However, our cytokine experiments suggest that patients with and without long-term proliferation show additional differences with regard to communication via the local cytokine network (i.e. IL-1/HGF/TNFα) with neighboring AML-supporting nonleukemic cells in their common bone marrow microenvironment.

In our opinion these previous observations strongly suggest that investigation of the contributions from single mediators to the ability of certain patients to show long-term in vitro proliferation would require extensive experiments and should rather be a separate study.

3.3 Samples used were from peripheral blood only. Would cells from bone marrow aspirates be different as this is the initial site of leukemic transformation?

From previous results, we would expect bone marrow cells to behave similarly to peripheral blood cells with inter-patient differences still being larger than differences between bone marrow and peripheral blood cells within a single patient. We added a paragraph to the discussion where we include previous studies on this matter, also from our research group (lines 399-411). We further added a paragraph to explain in more detail why we chose to work with peripheral blood blasts (lines 385-398).

To summarize, these two questions are now discussed in two new chapters of the Discussion section.

3.4 Are samples seeded on supporting cells similar to those not seeded on supporting cells?

This is now commented briefly in the Discussion section (page 13).

3.5 What is the initial cytokine profile of the samples? It would nice to see this data as this is what is occurring in a patient.

We included 55 of the cultures in another study where we collected initial (2-day culture) supernatants. Even though these cells were cultured in another medium and without endogenous cytokines, the samples are identical. When clustering the same 14 cytokines as in Figure 4, we got very good correlation between the patient distribution in two clustering analyses based on the alternative 2 days culture and the present 7 days cultures. This is now added in the text (lines 467-472).

Although the following comment has not been included in the present article, we also want to mention that we have published a study on cytokine secretion of AML patients previously (ref. 14); 31 of these patients overlapped with the 68 patients of the present study. Even though, we used different cell samples (ampullas) and another culture medium, we saw a high degree of correlation (p < 0.001) for subclassification of patients when comparing these previous results and the corresponding results from the present study. Thus, we see high reproducibility of cytokine profiles between different culture media, different samples of the same patients and different time intervals of short-term culture. Taken together these observations show that the cytokine release profiles of primary human AML cells shows a wide and reproducible variation between patients independent of the culture medium used and whether cells are cultured for 2 or 7 days.  

3.6 Hypothesis: “The capacity of long-term in vitro growth in the absence of AML-supporting non-leukemic cells is due to a different regulation of proliferation and a balance between pro- and antiapoptotic signaling that favor survival and proliferation even during suboptimal in vitro conditions.”

This statement has now been rewritten to describe our hypothesis more accurately, this part of the Introduction section now states that (page 2):

“……our hypothesis is that the ability of long-term in vitro growth in the absence of AML-supporting non-leukemic cells differs between patients; this capacity of long-term proliferation reflects a clinically relevant difference in the cellular ability to survive and proliferate under suboptimal conditions that is probably due to a different regulation of proliferation and/or balance between pro- and anti-apoptotic signaling.”

Section 2.4: This where an initial cytokine level before culture would be nice. Using the word association is good because there isn’t a direct causation shown but this would be nice to see by potentially blocking the cytokines present. Does this alter the long-term culture ability?

Section 2.6-2.8: Same as in 2.4.

In your hypothesis, it states that long term in vitro growth is due to “a different regulation of

proliferation and a balance between pro- and antiapoptotic signaling that favor survival and proliferation even during suboptimal in vitro conditions” but this is not actually proven. So, the hypothesis wasn't proven and additional experiments are needed. 

We agree that we did not prove the last part of our hypothesis, and this is now clearly stated in a new chapter in the last part of the Discussion section (page 15). As explained above it would probably require extensive new experiments to clarify the molecular mechanisms behind the differences in long-term proliferation between patients, e.g. the importance of various single cytokines within a panel of growth factors in individual patients. In our opinion this is outside the scope of the present article but rather needs to be addressed in a separate study. We hope our solutions are acceptable.

Round 2

Reviewer 2 Report

The discussion section is too long. The entire paper is too long, and it should be shortened.

Author Response

Thank you for your review of our manuscript.

We have undertaken following changes to our manuscript:

·        The Introduction section has been shortened from 464 to 403 words, corresponding to ca. 15% shortening based on the word count.

·        The Results section has not been shortened. The journal recommends detailed presentation of the results, and the two other reviewers have accepted the present presentation of our results after revision based on their own previous comments. For this reason the presentation of results has not been altered. We hope this is acceptable.

·        The Discussion section has been shortened. However, we have only shortened those parts of the Discussion that have not been edited throughout the first revision of the manuscript and which represent additional chapters that were required according to the comments from the two other reviewers to our original manuscript. We hope this decision can be accepted. The second revised version includes now 2759 words as compared to the 3065 words of the first revised version; corresponding to a shortening of 10% based on the word count. Finally, we would emphasize that this is a general shortening, we have not left out any whole chapters from the first Revised Version. We hope our solutions are acceptable.

·        We further stressed our main findings by reducing the Conclusion from 256 to 160 words (<30%).

We hope these changes are acceptable.